# Explorations on Thermodynamic and Kinetic Performances of Various Cationic Exchange Durations for Synthetic Clinoptilolite

**DOI:** 10.3390/molecules27082597

**Published:** 2022-04-18

**Authors:** Keling Wang, Bingying Jia, Yehong Li, Jihong Sun, Xia Wu

**Affiliations:** Beijing Key Laboratory for Green Catalysis and Separation, Department of Environmental and Chemical Engineering, Beijing University of Technology, Beijing 100124, China; wangkl@emails.bjut.edu.cn (K.W.); jiaby@bjut.edu.cn (B.J.); yehongli@emails.bjut.edu.cn (Y.L.)

**Keywords:** clinoptilolite, ion exchange, thermodynamic, kinetics, gas separation

## Abstract

Various cation–exchanged clinoptilolites (M–CPs, M = Li^+^, Cs^+^, Ca^2+^, Sr^2+^) were prepared, and their exchanged thermodynamic (and kinetic) properties and adsorption performances for CH_4_, N_2_, and CO_2_ were investigated. The results demonstrated that the relative crystallinity of M–CP_S_ decreased with the increase of exchange times. Their chemisorbed water weight loss gradually increased with the increasing exchange times, except that of Cs–x–CP. The Δ*_r_*$G_{m}^{\theta }$ values of exchange process of Li^+^, Cs^+^, Ca^2+^, or Sr^2^ presented the increased trend with the enhanced exchange times, but they decreased as the temperature increased. The negative Δ*_r_*$G_{m}^{\theta }$ values and the positive Δ*_r_*$H_{m}^{\theta }$ and Δ*_r_*$S_{m}^{\theta }$ values suggested that the exchanged procedure belonged to spontaneous, endothermic, and entropy-increasing behaviors; their kinetic performances followed a pseudo–second–order model. However, the calculated *E_a_* values of exchange process showed the increased tendencies with the enhanced exchange times, indicating that the exchange process became more difficult. Finally, the preliminary adsorption results indicated that the maximum adsorption amount at 273 K and 1 bar was 0.51 mmol/g of CH_4_ and 0.38 mmol/g of N_2_ by (Na, K)–CP, and 2.32 mmol/g of CO_2_ by Li–6–CP.

## 1. Introduction

Clinoptilolite (CP), as a member of the heulandite (HEU) group, belongs to monoclinic crystalline of porous aluminosilicates [1], whereas its framework system consists of a two–dimensional skeleton with three kinds of channels. The channel A (10-membered ring, 0.44 × 0.72 nm) parallels to channel B (8-membered ring, 0.40 × 0.55 nm) and to the c-axis of the unit cell, while channel C (8-membered ring, 0.41 × 0.40 nm) intersects channel A and B lying along the a-axis [2,3]. These channels are occupied by charge balancing cations and water molecules with a representative unit cell composition of (Na, K)_6_(Al_6_Si_30_O_72_)20H_2_O [4]. The cations distributed on the HEU skeletons are loosely held and can be exchanged to others [5], thus CP can be extensively used in a wide variety of adsorption, catalysts and environmental applications because of its chemical stability and high cationic exchange capacity. Osmanlioglu et al. [6] used Turkey’s natural CP as an ion exchanger to remove ^137^Cs, ^60^Co, ^90^Sr and ^110^Ag from liquid radioactive waste, showing high selective performances. Petrovic et al. [7] reported the migration of metalaxyl [N-(2,6-dimethylphenyl)-N-(methoxyacetyl) alanine methyl ester] to groundwater effectively reduced after adding CP to sandy soils, and Rehakova et al. [8] found that the concentration of polychlorinated biphenyls (PCBs) in plants decreased from 4765 to 2524 μg/kg by adding natural CP to PCBs-contaminated soil. Akgül [9] investigated the adsorption behaviors of Fe^3+^-grafted CP for the treatment of wastewater containing congo red dye, showing the highest adsorption capacity (up to 36.7 mg/g) due to the surface charge reversal and the formation of new hydroxyl groups. Stocker et al. [10] prepared modified natural CP by concentrated reagents in the NaCl (22.36%)-HCl (20%)-NaOH (32%) system and found that the adsorption capacity of CP for ammonium at lower concentrations could be enhanced via NaCl-treatment. Lin et al. [11] also found that NaCl-modified CP has high affinity to the ammonium in wastewater. Additionally, Kallo et al. [12] suggested that CP should be a potential catalyst for xylene and n-butene isomerization, methanol dehydration and acetylene hydration; Woo et al. [13] elucidated the synthetic CP for but-1-ene isomerization with 2-ethylpropene and found that the isobutene selectivity was as high as 93% for 1-butene conversion of 11%. The acidity and characteristic pore of CP structure seems to be the key of good performance. Khahlzadeh et al. [14] found that potassium hydroxide supported on natural CP is an efficient catalyst for the carboxymethylation of hemicellulose, because the CP with high adsorption and ion-exchange capacity has the positive effect of CP on KOH basicity.

On the other hand, the distinct microspore structure of CP before and after modifications allows the diffusion of smaller molecules such as CO_2_ and N_2_ with dynamic diameters of 0.34 nm and 0.36 nm, respectively, but impedes larger molecules such as CH_4_ with dynamic diameters of 0.38 nm [15]. Thus, CP is a promising adsorbent to separate the mixed gases composed of small molecular mixed gas by adjusting distributed position, located quantity and type of balanced cations on framework. For example, Hao et al. [16] reported that Na–CP and Ag–CP presented good equilibrium selectivity for N_2_ and CH_4_, showing the equilibrium separation factors up to 7.25 and 6.53 through pressure swing adsorption (PSA) at 0.2 MPa and 298 K, respectively. Meanwhile, the Na–CP is beneficial to promote the concentration of CH_4_ concentrated from 19.7 to 30.72%. Kouvelos et al. [17] demonstrated the ability of the Li–CP and Na–CP to separate N_2_–CH_4_ mixed gas through pressure swing adsorption (PSA) at 273 or 298 K, respectively. They found that all used CPs exhibited CH_4_ selective (selectivity values < 1) at 298 K, which is reversed at 273 K. The probable reason is due to an interaction between kinetics and thermodynamics. Recently, Kennedy et al. [18] reported the equilibrium adsorption and dynamic selectivity for CH_4_/N_2_ mixed gases using Ag^+^ exchanged CP as adsorbent. They concluded that the adsorption behavior of N_2_ is more likely to approach equilibrium than that of CH_4_ via thermodynamics and kinetic analysis; therefore, the N_2_/CH_4_ equilibrium selectivity value was higher, up to 7.6. More recently, Li et al. in our group [19] also demonstrated the selectivity for mixed gases and adsorption capacity of various cationic-exchanged CP, such as alkali metal cations (Li^+^, Na^+^, K^+^, Rb^+^, Cs^+^), alkaline metal cations (Mg^2+^, Ca^2+^, Sr^2+^, Ba^2+^), and transition metal cations (Fe^3+^, Ni^2+^, Cu^2+^, Co^2+^). Their selectivity for CH_4_-N_2_ mixed gas was strongly dependent on the cationic charge, size and position on HEU frameworks.

Despite the varieties of exchanged conditions and exchangeable cations used to modify CP for a wide range of potential applications, no comprehensive studies have been conducted to provide insights into the roles of adsorption and diffusion of cations occurring between liquid–solid and solid–solid interfaces. The cation exchange mechanism has often been described as very difficult, even by those who reported successful synthesis and modification of CPs [20,21,22]. The cation exchange performances of zeolites strongly depend upon Si/Al ratio and nature of the exchanged cations. Zamzow et al. found that the selectivity sequence of CPs was as follows, in order: Pb^2+^ > Cd^2+^ > Cs^+^> Cu^2+^ > Co^2+^ > Cr^3+^ > Zn^2+^ > Ni^2+^ > Hg^2+^ [23]. Subsequently, on the basis of Eisenman’s theory, Colella [24] successfully deduced the similar selection order: K^+^ > NH_4_^+^ > Na^+^ > Ca^2+^ > Mg^2+^. Besides, Ouki et al. [25] reported the selectivity of CP with the transition metal: Pb^2+^ > Cu^2+^ > Cd^2+^ > Zn^2+^ > Cr^3+^ > Co^2+^ > Ni^2+^. Li et al. [26] investigated the exchange process of the synthetic Na^+^–CP with Zn^2+^, Pb^2+^, Cd^2+^, and Cu^2+^. The results indicated that the exchange process follows the Langmuir model, pseudo–second–order kinetic equation. Additionally, they found that some cationic sites located on the open frameworks of CP can only be occupied by specific cations that are present in solution during the ion exchange process [26]. Argun [27] studied the effect of concentrations, contact time, and temperature to Ni^2+^–exchanged process of natural CP. The results showed that the process is spontaneous, exothermic, entropy reduction, accompanying with the pseudo–second–order kinetics and Langmuir isotherm. However, Pandey et al. [28] thought the ion-exchange process of MoS_2_–CP with Pb^2+^ under the conditions of MoS_2_–CP dose of 0.075 g, pH 6 at 328 K belonged to endothermic and entropy production. We [19] also explored the various cationic exchange behaviors at different temperature (313, 333, 353 K, respectively) under concentrations ranging 0.04 to 0.1 mol/L. The results showed that the cationic exchanged process was spontaneous, endothermic, and entropy increasing.

The objective of this work is to ascertain the exchange mechanism of various cations (Li^+^, Cs^+^, Ca^2+^, Sr^2+^) with synthetic CPs on the basis of our previous studies [19] and to further understand the thermodynamic–kinetic behaviors of the systematic exchanged procedures. For this purpose, a lot of experiments were accomplished establishing the relationships between exchanged capacity and exchanged times for various cations. In order to evaluate their capacities and selective separations in the removal of N_2_ or CO_2_ from CH_4_–conatined gas at 273 and 298 K for possible applications in efficient use of CH_4_ storage, the physicochemical and structural properties of resultant modified CPs were characterized via X-ray diffractive (XRD) patterns, thermal gravimetric analyzer (TG), scanning electron microscopy (SEM), Fourier transform infrared spectroscopy (FT-IR) spectra and N_2_ sorption isotherms. Besides, we obtained the thermodynamic parameters (enthalpy change (Δ*_r_*$H_{m}^{\theta }$), entropy change (Δ*_r_*$S_{m}^{\theta }$), and Gibbs energy change (Δ*_r_*$G_{m}^{\theta }$)) and kinetic parameters (apparent activation energy (*E_a_*)) of cation exchange process through the Gibbs–Helmholtz equation and Arrhenius formula. Finally, the adsorption performances of the various ion-exchanged CPs for N_2_, CO_2_, and CH_4_ were preliminarily explored.

## 2. Results and Discussion

### 2.1. Physical-Chemical Properties of before and after Ion-Exchanged CPs

The XRD patterns of before and after exchanged CPs are depicted in Figure 1 and Appendix A of the Electronic Supplementary Information (ESI) section—Appendix A. In Figure 1A, no major structural damage was observed after ion exchange of CP, as intensities of the main diffraction peaks for the HEU structures remained similar [29]. (020), (200) and (131) peak intensities of NH_4_–CP became stronger, but the intensities of other characteristic diffraction peaks remained nearly constant compared to that of (Na, K)–CP. Furthermore, the *d* values of (020), (200) and (131) were expanded to 0.8927, 0.7907 and 0.3976 nm, respectively (as shown in Appendix A). These results indicate the neat arrangement of NH_4_^+^ on the crystal planes. However, the decrease in intensity of the diffraction peaks was observed for the Cs^+^ and Sr^2+^ exchanged CPs (Figure 1A and Appendix A), which could be attributed to the high X-ray absorption of Cs^+^ and Sr^2+^, and the incorporation of these large cations led to the decrease in crystallinity [30]. Rodríguez-Iznaga et al. [31] also observed the variation of (020) peaks in the copper–silver bimetallic exchanged CP and believed that it was strongly linked to the extra framework cations positioned in the mirror plane perpendicular to *b*-axis in the HEU structures.

In addition, the relative crystallinity of the obtained CPs decreased with the enhanced exchange times (as seen in Table 1). In details, the relative crystallinity decreased from 1.542 for NH_4_–CP, to 1.165 for Li–6–CP, 0.399 for Cs–6–CP, 1.000 for Ca–4–CP, and 0.668 for Sr–8–CP, respectively. When exchanged time was the same (x = 2), the relative crystallinity decreased in the following order—Li–2–CP, Ca–2–CP, Sr–2–CP and Cs–2–CP—indicating that the effects of the cationic radius on the HEU structures were remarkable in the exchanged procedures. As is well known, the cationic radius of Li^+^, Ca^2+^, Sr^2+^ and Cs^+^ were 0.060, 0.099, 0.113, and 0.190 nm [32]. The low crystallinity of Cs–x–CP and Sr–x–CP also was because of the X-rays absorption by Cs^+^ and Sr^2+^ [30]. Similarly, the relative crystallinity order decreased as follows—Li–x–CP, Ca–x–CP, Sr–x–CP, and Cs–x–CP—when x > 2. Obviously, the crystallinity sequences of M–x–CPs should be dependent on the properties of exchange ions located on the frameworks and their interactions with the HEU structure rather than by a single factor.

As can be seen in Appendix A, the *d* spacing values of the characteristic diffractive peaks of NH_4_–CP presented the increased tendencies as compared with that of (Na, K)–CP. For example, *d* (020) value in NH_4_–CP was 0.8927 nm, larger than that (0.8909) in (Na, K)–CP. Obviously, this is because the radius of NH_4_^+^ in size of 0.143 nm is larger than that of Na^+^ and K^+^ [33]. The d spacing values of Cs–x–CP are largest among M–x–CPs, which was also attributed to the larger radius of the Cs^+^. Additionally, the positions of (020), (200) and (131) peaks of M–x–CPs shifted to small angle regions as compared with that of (Na, K)–CP, by accompanying with the increased *d* spacing values, indicating that the exchange procedures between cations and NH_4_^+^ mainly affect these crystal planes in the obtained CPs.

The FT-IR spectra of various M–x–CPs are shown in Appendix A. As can be seen, all samples presented almost the same profiles as the synthetic (Na, K)–CP, further indicating that the structure of the synthetic CP has not been destroyed after ion exchange. In details, the peak observed at approximately 3630 cm^−1^ belonged to the adsorbed water molecules, and the peak located at 1638 cm^−1^ corresponded to deformation vibrations of adsorbed water [34]. The peak at 1400 cm^−1^ was due to NH_4_^+^ exchanged in the CP and was linked to the NH (primary amino) bending vibration [35]. The obvious bands located at 1030 and 1208 cm^−1^ were associated to asymmetric stretching vibrations of the tetrahedral T-O-T (T = Si or Al) groups [36]. Others centered at 597 and 453 cm^−1^ were assigned to the outer tetrahedral double ring and the bending vibration of the T-O inside the tetrahedron, respectively [34].

Figure 1B and Appendix A present the TG profiles of various M–x–CPs. As can be seen, the weight loss process approximately could be divided into three stages, where the first stage shows a larger weight loss in the temperature range of 303–473 K due to the removal of loosely bound physisorbed water [37]. The second one in the higher temperature range of 473–673 K corresponded to the desorption of chemisorbed water or the decomposition of exchanged NH_4_^+^ [38]. The last procedure at 673–1073 K can be attributed to the dehydroxylation of CPs [39]. It is worth noting that the samples have almost the same weight loss rate (2.99~6.1%) in the first stage, while a big difference in the second (0.67~5.1%) and third stages (0.2~5%) appeared. These phenomena suggest that the used cations had no significant effect on the physisorbed water but obvious effect on the chemisorbed water [40]. 

Interestingly, the weight loss of Ca–2–CP and Ca–4–CP (Figure 1B) were 4.53% and 4.73% at 473–673 K, respectively, showing an increasing trend. Similarly, the weight loss of Li–x–CP, Ca–x–CP and Sr–x–CP (Appendix A) gradually increased, but that of Cs–x–CP gradually decreased with the enhanced exchange times in the temperature range 473–673 K, which may be related to cationic radius and their charge intensity. The radius of NH_4_^+^ is larger than that of Li^+^, Ca^2+^, and Sr^2+^, but smaller than that of Cs^+^ [32], while with the increase in cationic charge intensity and decrease in cationic radius, the stronger electric field in the skeleton cavity is more favorable to the adsorption and even coordination of water within the micropore channels. The weight loss of Ca–2–CP and Ca–4–CP is greater than that of (Na, K)–CP, but it is the opposite for Cs–2–CP and Cs–6–CP. The reason is that the Ca^2+^ radius is 0.099 nm, which is between that of Na^+^ and K^+^, but the Cs^+^ radius is larger than that of Na^+^ and K^+^ [32]. Obviously, the combination of cationic radius and their charge intensity determines the strength of the electric fields.

When the exchange capacity of Li^+^, Cs^+^, Ca^2+^ and Sr^2+^ with NH_4_^+^ reaches the maximum, the weight loss at 473–673 K decreased as follows—Li–6–CP > Ca–4–CP > Sr–8–CP > Cs–6–CP—which can also be interpreted by the effects of cationic radius (the smallest Li^+^) and the charge intensity (Ca^2+^ and Sr^2+^), while Cs^+^ is univalent and its radius is the largest among those cations.

The SEM images and elemental mappings of M–x–CPs are shown in Appendix A. As can be seen, the synthesized (Na, K)–CP exhibited the orderly stacked lamellar with smooth surface having size of around 2.5 μm, which is found in good agreement with the reported literature [41]. In comparison, other samples showed the almost same morphologies with stacked lamellar structures in size of around 2.0–2.5 μm. These observations indicated that the cationic exchange procedures had an unobvious impact on the morphology of CPs, proving that the intensity loss of their characteristic diffractive peaks (as shown in Figure 1A and Appendix A), particularly for Cs–x–CP (Figure 1A), are not caused by structural collapse. Meanwhile, the individually colored elemental mapping images showed that the elements such as O, Si, Al, K and Na were spread almost throughout (Na, K)–CP. 

### 2.2. Physical-Chemical Properties of before and after Ion-Exchanged CPs

Figure 2, Appendix A illustrated the influences of the exchanged parameters such as temperature and times on the exchange performances of Cs–x–CP, Ca–x–CP, Li–x–CP, and Sr–x–CP using NH_4_–CP as starting materials. As can be seen, the exchange capacity increased rapidly before 10 min and reached equilibrium roughly before 20 min. Similar phenomena also occurred when the exchanged temperature was enhanced from 313 to 353 K at the same exchange times. Lihareva et al. [42] investigated the exchange process of CP with Cs^+^ and Sr^2+^ at fixed pH 5.4 using Cs^+^ (or Sr^2+^)-contained solutions of 649 (or 200) mg/L, respectively. They also found the similar phenomena that the exchange process was mainly completed in the first few minutes and followed by a fast decay until the equilibrium was reached. Meanwhile, the increase in the exchange capacity at higher temperature may be due to the enhancement of mass transfer driving force via increasing exchange temperature, which is conducive to accelerate the cationic diffusion from aqueous solution to NH_4_–CP surfaces and enhance the accessibility of exchange site [43]. For example, the exchange capacity of Ca^2+^ was 19.5 mg/g at 313 K, and it increased to 23.4 and 24.7 mg/g at 333 and 353 K, respectively (as shown in Figure 2).

However, Figure 2 and Appendix A demonstrate that the exchange procedure of Cs^+^ and Li^+^ needed more times to reach the exchange equilibriums than that of Ca^2+^. The main reasons could be interpreted as follows. On the hand, the ionic radius of Cs^+^ (0.26 nm) is larger than that of Ca^2+^ (0.18 nm), which makes it difficult to enter the microporous channels and achieve exchange equilibrium [44]. On the other hand, Li^+^ ions occupy more exchanged sites in the HEU structures, although the effective ionic radius (0.076 nm) and hydrated radius (0.145 nm) of Li^+^ are lower than that of Ca^2+^. Meanwhile, it is likely that coordination of the Li^+^ with K^+^ results in occupying positions within the channels [44]. Additionally, the most exchange times were needed to reach equilibrium for Sr^2+^, which may be related to the exchanged sites in CP [42].

Accordingly, the thermodynamic equilibrium constants were evaluated [45], as follows.

Selective correction coefficient (*k_c_*):(1)$$k_{c}=\frac{A_{c}^{Z_{B}}\times {\left(1-A_{s}\right)}^{Z_{A}}}{{\left(1-A_{c}\right)}^{Z_{A}}\times A_{s}^{Z_{B}}}$$
where *A_s_* is the ratio of the concentration of exchange ions in solution to the initial concentration of exchange ions at equilibrium (*C/C*_0_). *A_c_* is the exchange ion ratio of the amount of exchange ion entering the phase of the used CP to the exchange ion (*q/Q*), *q* is the content of cation in CP at ion exchange equilibrium, *Q* is the maxim content of ammonia in CP. *Z_A_* and *Z_B_*, respectively, are the charges of the cations *A* and *B*, and the symbols ‘*c*’ and ‘*s*’ refer to the zeolite and solution phases, respectively.

Exchange equilibrium constant (*k_a_*):(2)$$ln\;k_{a}=\left(Z_{B}-Z_{A}\right)+\int_{0}^{1}lnk_{c}\;{dA}_{c}$$
*Z_A_* and *Z_B_* are the charges of cations *A* and *B*, respectively.

Exchange in thermodynamic function (Gibbs–Helmholtz formula):(3)$$ln\;k_{a}=\frac{\Delta_{r}H_{m}^{\theta }}{R}\times \frac{1}{T}+I$$
where *R* is Avogadro constant; *I* is a constant; according to Equation (3), there is a linear relationship between ln *k_a_* and *T*^−1^. Therefore, a fitting line can be obtained by plotting *T*^−1^ with *ln k_a_*. Then based on the slope of line *k*, Δ*_r_*$H_{m}^{\theta }$ value can be calculated.

The Δ*_r_*$G_{m}^{\theta }$ value of exchange process is calculated by the following equation:(4)$$\Delta_{r}G_{m}^{\theta }=-\frac{RTlnk_{a}}{Z_{B}\times Z_{A}}$$

Its Δ*_r_*$S_{m}^{\theta }$ value is calculated by the following equation:(5)$$\Delta_{r}S_{m}^{\theta }=\frac{\Delta_{r}H_{m}^{\theta }-\Delta_{r}G_{m}^{\theta }}{T}$$

Meanwhile, the kinetic parameters were calculated by Arrhenius Equation (6) [46], as follows:(6)$$k={Ae}^{-\frac{E_{a}}{RT}}$$
where *k* is the rate constant, *A* refers the frequency factor, *E_a_* denotes apparent activation energy, *R* and *T* are the gas constant and thermodynamic temperature, respectively.

On the basis of the relationships between *ln k_a_* and (*RT*)^−1^ × 10^4^ at different exchange times (Figure 3 and Appendix A), the positive Δ*_r_*$H_{m}^{\theta }$ values for all samples indicated that the cationic exchange process was endothermic (as shown in Table 2), in good agreement with our previous investigations [19] and Javanbakht et al.’s report [47]. Therefore, the exchange capacity at high temperature was larger than that at low temperature at the same exchange times. Barros et al. [48] also observed that the exchange process of K^+^ with Zeolite NaA was endothermic at 303 and 315 K; they explained that the positive values of Δ*_r_*$H_{m}^{\theta }$ of exchange process may be related to the dehydration of the hydrated in-going cations for achieving the most favorable exchange sites. Rodríguez-Iznaga et al. [35] found that Cu^2+^ originating from the dehydrated [Cu(H_2_O)_6_]^2+^ easily reach the exchangeable sites during the exchange of Cu^2+^ with NH_4_–CP.

However, Tarasevich and Polyakov [49] reported that the exchange process of K^+^, Cs^+^, and NH_4_^+^ with Na–CP was exothermic, but that of Li^+^ was endothermic at 299 K. They suggested that K^+^, Cs^+^ and NH_4_^+^ were located on the eight-membered ring of CP structure, while Li^+^ was distributed on the ten-membered ring. Obviously, the thermal values of exchange process are strongly dependent on the water stripping, location and species of the exchangeable cations.

As can be seen in Table 2, the calculated Δ*_r_*$G_{m}^{\theta }$ values for most of samples (except Li–6–CP) were negative, indicating that the exchange process was spontaneous, although their exchange procedures became gradually more difficult due to their increased *E_a_* values with the enhanced exchange times. Thus, we can conclude that the thermodynamics behavior is not dominant, affecting the ion exchange process between exchangeable cations and NH_4_–CP. 

As shown in Table 2, the negative Δ*_r_*$G_{m}^{\theta }$ values suggest that the ion exchange processes were feasible and spontaneous in nature. For all cation-exchanged CPs, the Δ*_r_*$G_{m}^{\theta }$ values were decreased with the increase of ion exchange temperature from 313 to 353 K. For example, Δ*_r_*$G_{m}^{\theta }$ value for Cs–6–CP (or Ca–4–CP) was around −0.025 (or −1.114) kJ/mol at 313 K, −0.181 (−1.225) kJ/mol at 333 K, and −0.416 (−0.581) kJ/mol. These results imply that the reasonable rising temperature is beneficial to promote the exchange process, due to accelerating the diffusion rate of exchangeable cations at high temperature. Pandey et al. also found a similar phenomenon [28].

The comparison results of the Δ*_r_*$G_{m}^{\theta }$ values (Table 2) suggested that the affinity of exchange procedure varied with various cations and different temperatures in the following order: Ca^2+^ > Sr^2+^ > Cs^+^ > Li^+^ at 313 K, while Ca^2+^ > Sr^2+^ > Li^+^ > Cs^+^ at 333 and 353 K. In general, all zeolites would prefer divalent instead of univalent cations because divalent cations with higher charges would better balance the negative network provided by an aluminum tetrahedral [48]. Thus, the cationic exchange behaviors of CP depend not only on the exchanged temperature but also cationic valence state.

Accordingly, the Δ*_r_*$S_{m}^{\theta }$ values were calculated using Gibbs–Helmholtz formula (as shown in mentioned above Equation (5)). As shown in Table 2, the exchange process of various cations belonged to an entropy enhancement, similar results were also described by Pandey et al. [28]. However, the present results were not consistent with those reported by Argun et al. [27] and Tarasevich et al. [49]. Particularly, Argun et al. [27] found the negative Δ*_r_*$S_{m}^{\theta }$ values during the removal of Ni^2+^ ion from aqueous solution using clinoptilolite as an adsorbent, which was mainly an Ni^2+^ adsorption process. In contrast, our work investigated the cation exchange behaviors with the exchangeable cations of clinoptilolite; therefore, the purpose of washing with deionized water in the cation exchange experiments was to remove the cations adsorbed on clinoptilolite. As demonstrated by Barros et al. [48], the cation adsorption process is usually exothermic, whereas the cation exchange behavior could be endothermic and entropic, increasing process. In our work, the exchange entropy involved the change in entropy of the solid and liquid phase of the system, therefore, the contributions to Δ*_r_*$S_{m}^{\theta }$ may arise from changes in water-cation environments, particularly, between the aqueous solution and clinoptilolites during the exchange process [49]. Additionally, an overall negative or positive change in entropy was also linked to the enthalpy values of the exchange process. 

Table 2 collects the calculated *E_a_* values of various cationic exchanged CPs. As can be seen, the increased *E_a_* values indicated that the exchange process needed more energy and became more difficult increased with the enhanced exchange times. This can be interpreted as follows. On the one hand, the exchange cations first located the easily exchangeable sites at initial exchange stages; on the other hand, the concentration difference of the exchanged cation distributed in between CP phase and liquid phase decreased, leading to the decrease of their adsorption capacity and diffusion performance in the CP surface and its pores. Specifically, our previous report [19] also demonstrated that the *E_a_* values also decreased with the enhancement of cationic concentration in solution because the improved concentration could be beneficial to mass transfer driving force. However, the *E_a_* values were around 6.2 to 12.6 kJ/mol, similar to Inglezakis’s results of 0.2–80 kJ/mol [50]. On the basis of the *E_a_* values, the order of cationic exchange was followed as Ca^2+^ > Li^+^ > Cs^+^ > Sr^2+^, which also provide a reason why the Sr^2+^ needed more exchange time to reach equilibrium. 

The pseudo-first-order kinetic (Equation (7)) and the pseudo-second–order kinetic (Equation (8)) models were used to explore the mechanism of exchange process for various cations at 353 K. The obtained results were shown in Figure 4, Appendix A and Table 3.
(7)$$ln\left(q_{e}-q_{t}\right)=lnq_{e}-k_{1}t$$
(8)$$\frac{t}{q_{t}}=\frac{1}{k_{2}\cdot q_{e}^{2}}+\frac{1}{q_{e}}t$$
where *q_e_* and *q_t_* are the amounts of exchanged uptake per mass of CP at equilibrium and corresponding time (min), respectively; *k*_1_ (min^−1^) is the rate constant of the pseudo–first–order model; and *k*_2_ (g/mg/min) is the rate constant of the pseudo–second–order model.

As can be seen in Figure 4 and Table 3, although the correlation coefficient (*R*^2^) of the pseudo–first–order dynamics linear fitting data was higher than 0.8, the theoretical *q_e_* values were much lower than the experimental values. In contrast, the pseudo–second–order kinetics model (*R*^2^ > 0.999) provided a near–perfect match between the theoretical and experimental values. Comparing the correlation coefficient and the fitting equilibrium exchanged amount, we found the exchange process of various M–x–CPs well-fitted pseudo–second–order kinetics, suggesting that the chemical kinetic performance plays an important role in the whole exchange process between various used cations and NH_4_–CP. 

According to Rodríguez-Iznaga et al.’s demonstrations [43], the ion exchange process could be divided into four stages, as follow: (1) the diffusions between exchanged cations dispersed in the solution and the liquid system; (2) the transportations of the exchanged cations surrounding the CP through the CP surfaces; (3) the movements of the exchanged cations into the exchangeable positions; (4) the occurrences of the cationic exchange behaviors at the exchangeable sites of CP. The first and second steps belong to the external diffusion, while the third and fourth steps correspond to intra-crystalline diffusion.

Additionally, Table 3 presented that the increased *k*_2_ values indicated that the equilibrium state was reached more quickly with the enhancement of the exchange times, similar to the calculated results by Ijagbemi et al. [51]. The differences of the *k*_2_ values among the various cationic exchanged CP may be related to the valence charge and radius of the exchanged cations.

### 2.3. CH_4_, N_2_ and CO_2_ Adsorption Performances of Various Ion-Exchanged CPs

Figure 5 and Appendix A depicted the CO_2_, N_2_ and CH_4_ adsorption isotherms of various cationic-exchanged CPs at 273 K and 298 K, respectively. As can be seen, the CH_4_, N_2_ and CO_2_ uptake of samples at 273 K are all higher than that at 298 K. The maximum adsorption amount was 0.51 mmol/g of CH_4_ and 0.38 mmol/g of N_2_ by (Na, K)–CP (Figure 5A) and 2.32 mmol/g of CO_2_ by Li–6–CP (Figure 5B) at 273 K and 1 bar. Obviously, the observed adsorption capacities followed the expected tendencies of CO_2_ > CH_4_ > N_2_, which can be well interpreted in terms of the polarizability order, as follows: CO_2_ > CH_4_ > N_2_. The larger polarizability means that gas molecule has a greater tendency to generate the instantaneous dipoles with the surfaces, resulting in greater electrostatic potential and allowing greater surface occupancy [52]. Besides that, the adsorption capacity of CO_2_ is largest among these gases due to its lower kinetic hindrances than CH_4_ and N_2_. 

Additionally, the CO_2_ and CH_4_ adsorption capacity at 273 K followed the order of Li–6–CP > Cs–6–CP > Sr–8–CP > Ca–2–CP, while the order of N_2_ adsorption capacity became as follows: Li–6–CP > Sr–8–CP ≈ Cs–6–CP > Ca–2–CP. Obviously, the Li–6–CP presented the highest adsorption capacity for all gases; one of the main reasons is that Li^+^ tends to occupy the smallest space in the HEU channels, which is conducive to opening more molecules adsorption space. Comparably, the Ca–4–CP exhibited low adsorption capacity for all gases because the occupations of the intersections sites of the channels A and B easily result in a more effective blocking of the microporous networks [17]. Although Kennedy et al. [44] reported that channel A in Cs–6–CP was completely open, which was useful to the entry of the gas into the inner channels, Sacerdoti et al. [53] further demonstrated that Sr^2+^ distributed in Sr–CP preferentially occupied the regions closer to the side of the channels, rather than its middle sites; the higher cavity volume makes it relatively easy for gases to enter; the large radius of Cs^+^ and Sr^2+^ limits their maxim adsorption capacity. It is obvious that the adsorption capacity of the (Na, K)–CPs as the raw materials should be significantly affected after different ions exchange, because of the differences of radius and the sites of cations in the HEU frameworks.

The calculated adsorption heat of gases could be calculated on the basis of the Clausius–Clapeyron equation (Equation (9)):
(9)$$ln\frac{P_{1}}{P_{2}}=\frac{\Delta H}{R}(\frac{1}{T_{2}}-\frac{1}{T_{1}})$$
where *P*_1_ and *P*_2_ are the relative pressures under the temperatures of *T*_1_ and *T*_2_, ∆*H* is isosteric adsorption heat.

To further understand the adsorption performances, the adsorption heat of CH_4_, N_2_ and CO_2_ were assessed according to the Clausius–Claperyron equation (as shown in mentioned above Equation (9)). As can be seen in Figure 6 and Appendix A, the adsorption heat values of all samples were negative for all gases, meaning that the adsorption process was exothermic. Meanwhile, the adsorption heat of CO_2_, N_2_ and CH_4_ for (Na, K)–CP, Cs–6–CP, Ca–4–CP and Sr–8–CP presented decreased tendency with increased absorbed amount, but it is opposite for Li–6–CP. Zhang et al. [54] also observed the similar results for CO_2_ adsorption using LiCHA zeolite as adsorbent, which could be explained by the strong adsorbate–adsorbate interactions. Compared with (Na, K)–CP, Cs–6–CP and Ca–4–CP showed higher isosteric adsorption heats for CO_2_, indicating that the Cs^+^ and Ca^2+^ exchange can enhance the interaction with CO_2_ and adsorbents. For Cs–6–CP, the absolute value of the isosteric heat of CO_2_ adsorption is around 45 kJ/mol, indicating that it can be regarded as a physical absorption process. These results implied that the Cs^+^ exists mainly as charge balancing cation rather than hydroxide. As shown in Figure 6A,C, the adsorption heat of CO_2_, CH_4_ and N_2_ showed a similar tendency with the increase of absorbed amount for (Na, K)–CP and Ca–4–CP. However, the adsorption heat of CO_2_ and CH_4_ increased, but that of N_2_ decreased with the increase in their absorbed amount for Ca–4–CP, implying a stronger adsorption affinity between Ca–4–CP and CO_2_ (or CH_4_), which is related to the exchanged cationic property and the polarizability of adsorbate molecules [55,56].

## 3. Materials and Methods

### 3.1. Materials

NaOH (99% of purity), Al(OH)_3_ (82%) supplied by Beijing Chemical Works. KOH (96%), NH_4_Cl (99%), LiCl (95%), CsCl (99%), CaCl_2_ (99%), SrCl_2_ (99%) were provided by Tianjin Fuchen Chemical Reagents factory. Ludox (1.2 g/mL, 30%) was purchased from Qingdao Ocean Chemical Co., Ltd. All the chemicals were of A.R. grade. Deionized water with a resistivity of 18.25 MΩ cm at 298 K was produced by Zhiang-Best Water Purifier. The standard solutions of Li^+^, Cs^+^, Ca^2+^, Sr^2+^, Si^4+^ and Al^3+^ were provided by Shanghai Aladdin Reagents factory and their concentrations were 10,000, 10,000, 1000, 10,000, 1000 and 1000 ug/mL, respectively.

### 3.2. Synthesis of CP and Procedure of Ion Exchange

A certain amount of NaOH, KOH, Al(OH)_3_ and deionized water (molar ratio:1.39 Na_2_O: 1.39 K_2_O: 1 Al_2_O_3_: 296 H_2_O) were mixed and stirred at 423 K for 3 h to obtain a clarified aluminate solution. Next, 75 mL of deionized water, 44 mL of silica sol and 10 wt% natural CP (screened by 400 mesh sieve) were added to the prepared aluminate solution, and the mixture was stirred at room temperature for 2 h. Subsequently, the mixed solution was crystallized in a Teflon-lined autoclave at 423 K for 72 h. After that, the obtained product was washed with 1000 mL deionized water and dried in a drying oven at 393 K for 12 h. The synthesized CP is named as (Na, K)–CP (as-synthetic CP).

Before exchanging with metal ion, the synthesized (Na, K)–CP was converted to NH_4_–CP by ion exchange with aqueous 1 mol/L NH_4_Cl solution. Two grams of (Na, K)–CP was added to 200 mL of NH_4_Cl solution and stirred for 2 h at 353 K. Then, the sample was washed with deionized water until no Cl^−^ was detected in the wash water via 0.1 mol/L AgNO_3_ solution test, and dried at 393 K for 12 h. This procedure was repeated ten times to obtain the NH_4_^+^–exchanged CP. The content of ammonium ion in NH_4_–CP is expressed by the difference between the content of sodium and potassium ion in (Na, K)–CP and NH_4_–CP.

After the process of converting the exchangeable cations in (Na, K)–CP to NH_4_^+^, NH_4_–CP was used as raw material to react with 0.2 mol/L cation salt solutions for 1, 2, 3, 5, 7, 10, 20 and 30 min at 313, 333 and 353 K, respectively. The solid:liquid ratio was 1 g of solid and 200 mL of solution. Following this step, the samples were washed with deionized water until no Cl^-^ was detected, then dried at 393 K for 12 h. The sample for 30 min in the first exchange process was used as the raw material for the second exchange process, and the same steps were repeated until the content of cation in CP was no longer increasing. The resultant samples at 353 K were labeled as M–x–CP (M = Li^+^, Cs^+^, Ca^2+^, Sr^2+^, and x = the exchange times). On the basis of preliminary work of the research group, the study of kinetic and thermodynamics starts from the second exchange process. The experiments were repeated three times and the obtained data were averaged.

The cationic exchange capacity (*q*) was calculated using the following Equation (10):(10)$$q\;(\mathrm{mg}/g)=\frac{c\times v}{m}$$
where *m* is the mass of CP (g), *c* refers to the cation concentration in solution (mg/L), *v* denotes the volume of the solution (L).

### 3.3. Characterizations

The structures of the synthesized (Na, K)–CP and ion-exchanged CPs were analyzed by Bruker AXS D8 Advance X-ray diffractometer using Cu Kα X-ray source in the range of 5–50° at 35 kV and 20 mA. The relative crystallinity of various CPs was calculated from the sum of intensities of ten main characteristic diffractive peaks, including (020), (200), (111), (13-1), (131), (22-2), (24-2), (151), (530), and (061), while the crystallinity of (Na, K)–CP was set to 1. The FT-IR spectroscopy was measured by Nicolet 6700 analyzer in the wave number range of 400–4000 cm^−1^. The TG profiles were conducted by Perkin–Elmer Simultaneous Thermal Analyzer in following N_2_ using a heating rate of 10 K min^−1^ from 303 to 1173 K. The SEM images were recorded at 15 kV using a JEOLJEM-220. A trace sample was taken into a 1 mL centrifuge tube, followed by adding 1 mL ethanol, and the powder was evenly dispersed into ethanol by use of ultrasound, and then a drop of liquid was dropped onto the conductive adhesive on the sample table; finally, platinum was uniformly sprayed on the sample surface for 1 min.

The metal ion content of the obtained samples was measured by atomic absorption Spectrometer (AAS) (PerkinElmer Optima 2000 DV), while their Si/Al molar ratio and the content of Cs^+^ were obtained on the basis of inductively coupled plasma (ICP) (PerkinElmer Optima 8300) data. The standard curves of Li^+^ (or Ca^2+^, Sr^2+^, Cs^+^, Si^4+^, Al^3+^) were shown in Appendix A, their correlation coefficients (*R*^2^) all were higher than 0.999.

The pre-treatment process of test samples was as follows: a certain mass of the samples obtained before and after ion exchange was weighed to a plastic tube, then 2–3 drops of HF aqueous solution (23 mol/L) and a certain amount of HNO_3_ solution (volume fraction 2%) were dropped to completely dissolve it. Next, the solution was transferred to a 100 mL volumetric flask, which could be used for various cationic determinations. 

The chemical formulas of various CPs were obtained as follows: the contents of various cations were obtained on the basis of AAS and ICP data. The chemical formulas were calculated on the molar ratio of each atom to aluminum atom while the molar number of oxygen atom was 72. 

The CH_4_, N_2_ and CO_2_ adsorption isotherms of various samples were measured using a JWGB (JW-BK 300, Beijing) at 273 K and 298 K, respectively. The measures procedures were as follows: 0.2 g of sample was added to the sample tube and preheated for 6 h at 393K under vacuum to remove water and gas. Next, CH_4_, N_2_ or CO_2_ adsorption tests were performed at constant temperatures (273 and 298 K) until the gas pressure (CH_4_, N_2_ or CO_2_) was increased about 1.0 bar.

## 4. Conclusions

The influences of the exchanged parameters for different cations (Li^+^, Cs^+^, Ca^2+^, Sr^2+^) with CP on the structure properties and adsorption performances of the ion-exchanged CPs were investigated via various characterizations. The results showed that the relative crystallinity and properties of adsorbed water of the ion-exchanged CPs were strongly dependent on the size and charge of the used cations, although the HEU structures of before and after ion-exchanged CPs remained intact. The thermodynamic–kinetic parameters (Δ*_r_*$G_{m}^{\theta }$, Δ*_r_*$H_{m}^{\theta }$, Δ*_r_*$S_{m}^{\theta }$, *E_a_*) demonstrated that the exchanged behavior belonged to a spontaneous, entropic and endothermic procedure, while the Δ*_r_*$G_{m}^{\theta }$ and *E_a_* values all enhanced with the increased exchange times, suggesting that the cationic exchange procedures became more difficult. We also found that the ion exchange process was mainly kinetic-dominant and presented a near perfect match with pseudo–second–order model. The maxim exchange capacity of Li^+^, Cs^+^, Ca^2+^ and Sr^2+^ are 13.3, 252.7, 36.4 and 82.7 mg/g, respectively. The adsorption capacity of the resultant CPs showed that the raw material has the higher adsorption capacity (up to 0.51, 0.38 and 1.74 mmol/g for CH_4_, N_2_ and CO_2_ respectively). 

## Figures and Tables

**Figure 1 molecules-27-02597-f001:**
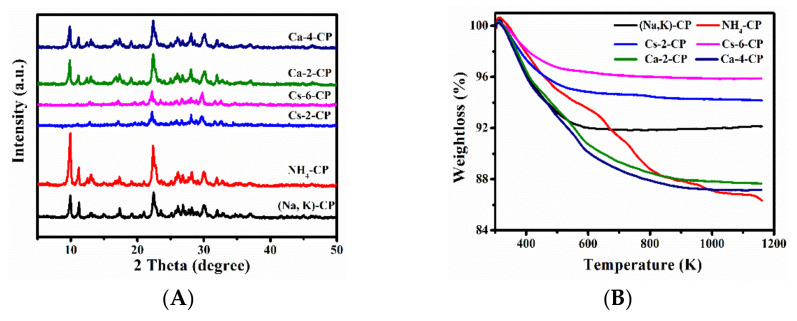
XRD patterns (**A**) and TG profiles (**B**) of (Na, K)–CP of various ion-exchanged CPs.

**Figure 2 molecules-27-02597-f002:**
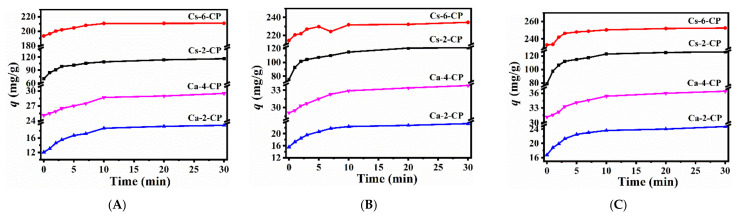
Exchanged kinetic behaviors of Cs–x–CP and Ca–x–CP at 313 K (**A**), 333 K (**B**), and 353 K (**C**).

**Figure 3 molecules-27-02597-f003:**
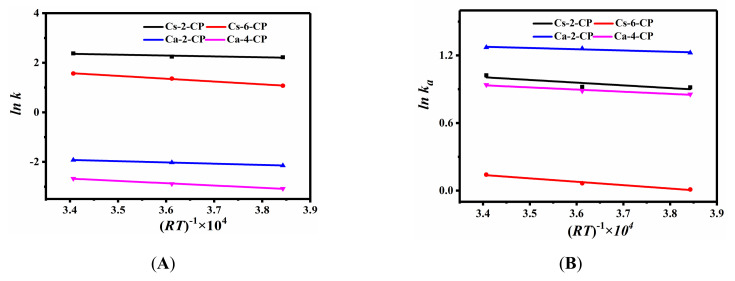
Relationships between *ln k* and (*RT*)^−1^ × 10^4^ (**A**), as well as *ln k_a_* and (*RT*)^−1^ × 10^4^ (**B**).

**Figure 4 molecules-27-02597-f004:**
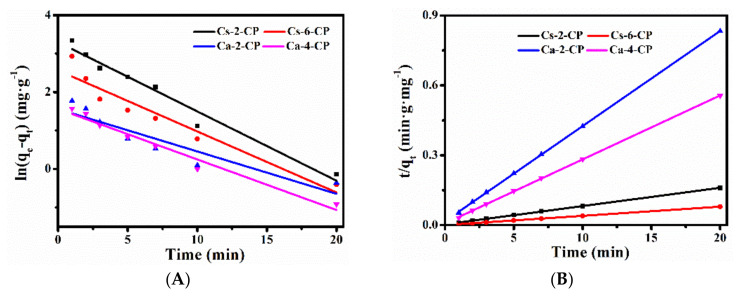
Pseudo-first-order model (**A**) and Pseudo-second-order model (**B**) fitted for Cs–x–CP and Ca–x–CP at 353 K.

**Figure 5 molecules-27-02597-f005:**
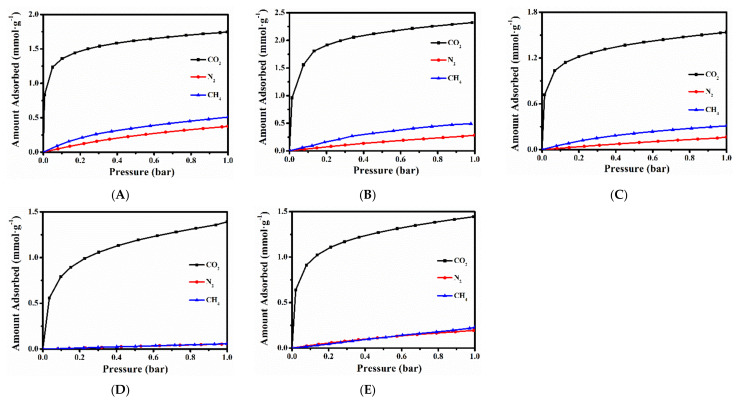
Adsorbed isotherms of (Na, K)–CP (**A**), Li–6–CP (**B**), Cs–6–CP (**C**), Ca–4–CP (**D**), and Sr–8–CP (**E**) using CO_2_, N_2_ and CH_4_ as adsorbate at 273 K.

**Figure 6 molecules-27-02597-f006:**
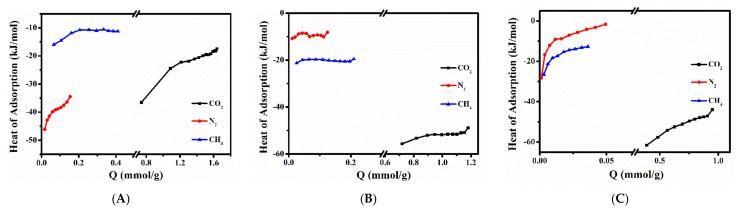
Adsorption heat of CO_2_, N_2_ and CH_4_ of (Na, K)–CP (**A**), Cs–6–CP (**B**), and Ca–4–CP (**C**).

**Table 1 molecules-27-02597-t001:** Collections of chemical structure formula and relative crystallinity of various CPs.

Samples	Chemical Formula	Relative Crystallinity
(Na, K)–CP	Na_1.122_K_6.433_Si_28.387_Al_7.631_O_72_	1.00
NH_4_–CP	Na_0.147_K_0.544_(NH_4_)_6.864_Si_28.387_Al_7.631_O_72_	1.542
Li–2–CP	Na_0.147_K_0.544_(NH_4_)_3.869_Li_2.995_Si_28.387_Al_7.631_O_72_	1.387
Li–3–CP	Na_0.147_K_0.544_(NH_4_)_2.683_Li_4.181_Si_28.387_Al_7.631_O_72_	1.234
Li–4–CP	Na_0.147_K_0.544_(NH_4_)_1.671_Li_5.193_Si_28.387_Al_7.631_O_72_	1.233
Li–5–CP	Na_0.147_K_0.544_(NH_4_)_1.145_Li_5.719_Si_28.387_Al_7.631_O_72_	1.191
Li–6–CP	Na_0.147_K_0.544_(NH_4_)_0.679_Li_6.185_Si_28.387_Al_7.631_O_72_	1.165
Cs–2–CP	Na_0.147_K_0.544_(NH_4_)_3.814_Cs_3.050_Si_28.387_Al_7.631_O_72_	0.495
Cs–3–CP	Na_0.147_K_0.544_(NH_4_)_2.708_Cs_4.156_Si_28.387_Al_7.631_O_72_	0.454
Cs–4–CP	Na_0.147_K_0.544_(NH_4_)_1.791_Cs_5.073_Si_28.387_Al_7.631_O_72_	0.446
Cs–5–CP	Na_0.147_K_0.544_(NH_4_)_1.214_Cs_5.650_Si_28.387_Al_7.631_O_72_	0.430
Cs–6–CP	Na_0.147_K_0.544_(NH_4_)_0.744_Cs_6.120_Si_28.387_Al_7.631_O_72_	0.399
Ca–2–CP	Na_0.147_K_0.544_(NH_4_)_2.894_Ca_1.985_Si_28.387_Al_7.631_O_72_	1.132
Ca–3–CP	Na_0.147_K_0.544_(NH_4_)_1.850_Ca_2.507_Si_28.387_Al_7.631_O_72_	1.061
Ca–4–CP	Na_0.147_K_0.544_(NH_4_)_1.014_Ca_2.925_Si_28.387_Al_7.631_O_72_	1.000
Sr–2–CP	Na_0.147_K_0.544_(NH_4_)_3.908_Sr_1.478_Si_28.387_Al_7.631_O_72_	0.800
Sr–4–CP	Na_0.147_K_0.544_(NH_4_)_2.350_Sr_2.257_Si_28.387_Al_7.631_O_72_	0.782
Sr–6–CP	Na_0.147_K_0.544_(NH_4_)_1.374_Sr_2.745_Si_28.387_Al_7.631_O_72_	0.772
Sr–8–CP	Na_0.147_K_0.544_(NH_4_)_0.786_Sr_3.039_Si_28.387_Al_7.631_O_72_	0.668

**Table 2 molecules-27-02597-t002:** Summaries for Δ*_r_*$G_{m}^{\theta }$ (kJ/mol), Δ*_r_*$S_{m}^{\theta }$ (J/mol k), Δ*_r_*$H_{m}^{\theta }$ (kJ/mol), and *E_a_* (kJ) values of various cationic exchange process with NH_4_–CP.

Samples	Exchange Times	313 K		333 K		353 K		Δ*_r_*$H_{m}^{\theta }$	*E_a_*
		Δ*_r_*$G_{m}^{\theta }$	Δ*_r_*$S_{m}^{\theta }$	Δ*_r_*$G_{m}^{\theta }$	Δ*_r_*$S_{m}^{\theta }$	Δ*_r_*$G_{m}^{\theta }$	Δ*_r_*$S_{m}^{\theta }$		
Li–x–CP	2	−4.603	30.59	−4.897	29.64	−5.796	30.50	4.976	6.220
	3	−3.226	22.69	−3.602	22.45	−4.138	22.70	3.878	7.349
	4	−2.934	23.16	−3.513	23.50	−3.857	23.14	4.316	7.839
	5	−2.390	14.39	−2.631	14.25	−2.968	14.40	2.117	8.380
	6	−2.244	15.61	−2.609	15.76	−2.863	15.59	2.643	9.319
Cs–x–CP	2	−3.479	14.99	−3.706	14.77	−4.081	14.99	1.214	3.504
	3	−2.749	30.08	−3.482	30.48	−3.949	30.08	6.672	6.406
	4	−2.694	23.12	−3.105	22.97	−3.623	23.13	4.547	8.230
	5	−2.442	12.39	−2.581	12.07	−2.823	12.07	1.439	10.373
	6	−2.273	13.22	−2.497	13.10	−2.804	13.23	1.867	11.370
Ca–x–CP	2	−1.593	8.641	−1.749	8.591	−1.868	8.441	1.113	5.082
	3	−1.217	14.16	−1.406	13.87	−1.578	13.57	3.216	8.866
	4	−1.114	9.663	−1.225	9.416	−1.380	9.322	1.912	9.283
Sr–x–CP	2	−1.431	24.14	−1.629	23.28	−2.009	23.04	6.128	5.308
	4	−0.968	11.71	−1.116	11.46	−1.265	11.23	2.701	6.409
	6	−0.803	9.976	−0.912	9.703	−1.055	9.560	2.321	9.836
	8	−0.548	5.109	−0.602	4.965	−0.687	4.924	1.052	12.604

**Table 3 molecules-27-02597-t003:** Summaries for the kinetic constants of various cationic exchange CPs.

Samples	Exchange Times	Pseudo-First-Order			Pseudo-Second-Order			*q_e_* (*exp*) (mg·g^−1^)
		*k*_1_ (min^−1^)	*R* ^2^	*q_e_* (*cal*) (mg·g^−1^)	*k*_2_ (g·mg^−1^·min^−1^)	*R* ^2^	*q_e_* (*cal*) (mg·g^−1^)	
Li–x–CP	2	0.2003	0.8236	1.075	0.4291	0.9998	6.557	6.46
	3	0.1567	0.9978	1.864	0.2556	0.9996	9.066	9.01
	4	0.1289	0.9470	1.610	0.3004	0.9999	11.186	11.19
	5	0.2853	0.9889	1.531	0.4800	0.9999	12.422	12.33
	6	0.1568	0.9003	0.744	0.7013	0.999	13.351	13.33
Cs–x–CP	2	0.1801	0.9712	27.067	0.0179	0.9998	128.205	125.87
	3	0.1131	0.8677	29.818	0.0158	0.9999	169.492	171.53
	4	0.2169	0.9788	24.039	0.0245	0.9999	212.766	209.38
	5	0.0954	0.8121	14.189	0.0462	0.9999	232.56	233.18
	6	0.1666	0.9286	14.326	0.0400	0.9999	250.00	252.57
Ca–x–CP	2	0.1101	0.8656	4.754	0.0991	0.9999	24.510	24.70
	3	0.0903	0.9237	5.325	0.0857	0.9994	30.675	31.20
	4	0.1315	0.9716	4.779	0.1022	0.9999	36.364	36.40
Sr–x–CP	2	0.2217	0.9177	10.746	0.0434	0.9996	41.322	40.20
	4	0.1716	0.9412	7.039	0.0729	0.9999	61.728	61.40
	6	0.197	0.9542	4.123	0.1474	0.9999	75.188	74.70
	8	0.1127	0.8494	2.188	0.2440	0.9999	82.644	82.70

## Data Availability

Not applicable.

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
