# Peer review of "Explorations on Thermodynamic and Kinetic Performances of Various Cationic Exchange Durations for Synthetic Clinoptilolite"

_molecules, 2022, doi:10.3390/molecules27082597_

Round 1
Reviewer 1 Report
It was an interesting article.
- Article by topic: Explorations on thermodynamic and kinetic performances of various cationic exchange durations for synthetic clinoptilolite was an interesting and valuable article.
- - The abstract of the article has well shown all the results related to this research
- - In the introduction, the importance of Clinoptilolite and its properties are well expressed. The importance of this material in water and wastewater treatment processes and its use as an adsorbent has been stated.
- Selective ion exchange, which is important in water and wastewater treatment and is well performed by this substance, has been correctly expressed by reference to related studies.
- - The chemical structure of different Clinoptilolites is well compared in the results.
- - Exchanged kinetic behaviors of Cs-x-CP and Ca-x-CP at 313 K, 333 K and 353 K, show the importance of the authors to the difference of this property in different types of clinoptylates.
- - The effect of different temperatures on the structure and function of Clinoptilolite has been studied and well reported.
- The kinetic constants of different cation exchange have been investigated.
- - The materials and methods section should be moved with the results.
- - How to synthesize and perform experiments is generally written and seems to be enough as a scientific article.
Author Response
Reviewer #1:
It was an interesting article.
Response: We thank the reviewer for the positive comments.
Comments 1-01: Article by topic: Explorations on thermodynamic and kinetic performances of various cationic exchange durations for synthetic clinoptilolite was an interesting and valuable article.
Response: We thank the reviewer for the positive comments.
Comments 1-02: The abstract of the article has well shown all the results related to this research.
Response: We thank the reviewer for the positive comments.
Comments 1-03: In the introduction, the importance of Clinoptilolite and its properties are well expressed. The importance of this material in water and wastewater treatment processes and its use as an adsorbent has been stated.
Response: We thank the reviewer for the positive comments.
Comments 1-04: Selective ion exchange, which is important in water and wastewater treatment and is well performed by this substance, has been correctly expressed by reference to related studies.
Response: We thank the reviewer for the positive comments.
Comments 1-05: The chemical structure of different Clinoptilolites is well compared in the results.
Response: We thank the reviewer for the positive comments.
Comments 1-06: Exchanged kinetic behaviors of Cs-x-CP and Ca-x-CP at 313 K, 333 K and 353 K, show the importance of the authors to the difference of this property in different types of clinoptylates.
Response: We thank the reviewer for the positive comments.
Comments 1-07: The effect of different temperatures on the structure and function of Clinoptilolite has been studied and well reported.
Response: We thank the reviewer for the positive comments.
Comments 1-08: The kinetic constants of different cation exchange have been investigated.
Response: We thank the reviewer for the positive comments.
Comments 1-09: The materials and methods section should be moved with the results.
Response: We thank the reviewer for the positive comments. However, the materials and methods section were formatted according to the Author Guider of the Molecules.
Comments 1-08: How to synthesize and perform experiments is generally written and seems to be enough as a scientific article.
Response: We thank the reviewer for the positive comments.

Reviewer 2 Report
The authors studied ion exchange and gas separation on clinoptilolite. It is an interesting paper with many data, but some improvement is needed.
- line 84-88: what is "exchange difficulty"?
- 2; line 94: open-skeleton? alien cation?
- 122-135: the authors must be aware that peak intensities and peak shapes depends on many parameters, not only the type of exchangeable cation.
- line 163: the IR bands are not due to bridging OH molecules. 3607 and 1640 are the bands of adsorbed water; the 2981 band is probbably a C-H stretch of carbonaceous impurities.
- line 228-9: it is the hydration radius of Li; for Cs it is the size of the cation even without hydration water.
- eq. 1 is strange to me and needs explanation. Is it a general expressions valid for exchange of univalent cations and divalent cations?
- eq. 3: The constant I is in my opinion the entropy term.
- line 284: improved temperature: what is it?
Author Response
Reviewer #2:
The authors studied ion exchange and gas separation on clinoptilolite. It is an interesting paper with many data, but some improvement is needed.
Response: We thank the reviewer for the useful and positive comments. We have revised the manuscript carefully according to your constructive suggestions.
Comments 2-01: line 84-88: what is "exchange difficulty"?
Response 2-01: We thank the reviewer’s useful suggestions. “exchange difficulty” was revised “the selectivity sequence”.
The cation exchange performances of zeolites strongly depend upon Si/Al ratio and nature of the exchanged cations. Zamzow et al found that the selectivity sequence of clinoptilolite was as follows in order: Pb2+ > Cd2+ > Cs+> Cu2+ > Co2+ > Cr3+ > Zn2+ > Ni2+ > Hg2+ [1].
Appendix references
[1] M. J. Zamzow, B. R. Eichbaum, K. R. Sandgren, D. E. Shanks. Removal of Heavy Metals and Other Cations from Wastewater Using Zeolites. Sep. Sci. 1990, 25, 1555-1569.
The above description was added in lines 84-87 of the revised manuscript, as follow:
The cation exchange performances of zeolites strongly depend upon Si/Al ratio and nature of the exchanged cations. Zamzow et al found that the selectivity sequence of clinoptilolite was as follows in order: Pb2+ > Cd2+ > Cs+> Cu2+ > Co2+ > Cr3+ > Zn2+ > Ni2+ > Hg2+ [23].
Comments 2-02: line 94: open-skeleton? alien cation?
Response 2-02: We thank the reviewer’s useful comments. “open-skeleton” was changed into “open-frameworks” in line 93 of the revised manuscript.
“alien cation” means certain type of cations that is present in solution during the ion exchange process [2], which was changed in line 94 of the revised manuscript.
Appendix references
[2] Y.R. Li, P. Bai, Y. Yan, W. F. Yan, W. Shi, R. R. Xu. Removal of Zn2+, Pb2+, Cd2+, and Cu2+ from aqueous solution by synthetic clinoptilolite. Micro. Meso. Mater. 2019, 273, 203-211.
Comments 2-03: 122-135: the authors must be aware that peak intensities and peak shapes depends on many parameters, not only the type of exchangeable cation.
Response 2-03: We thank the reviewer’s useful comments. You are right, the peak intensities and peak shapes of XRD patterns mainly are dependent on the extent of crystallinity of the particular plane, interplanar spacing, crystallite size, doping and vacancy. Generally, the changes in peak intensity are related to the arrangements of some atomic positions within the unit cell and/or an atomic density change [3], while, their shapes are associated to the simultaneous effects from a reduced size of the microcrystals (crystallites) and/or an increased amount of lattice disorder [4].
In Figure 1A, no major structural damage was observed after ion-exchange of CP, as intensities of the main diffraction peaks for the HEU structures remained similar [5]. (020), (200) and (131) peaks intensities of NH4-CP became stronger, but the intensities of other characteristic diffraction peaks remained nearly constant compared with that of (Na, K)-CP. Furthermore, the d values of (020), (200) and (131) were expanded to 0.8927, 0.7907 and 0.3976 nm, respectively (as shown in Table S1). These results indicate the neatly arrangement of NH4+ on the crystal planes. However, the decrease in intensity of the diffraction peaks was observed for the Cs+ and Sr2+ exchanged CPs (Figure 1A and Figure S1B), which could be attributed to the high X-ray absorption of Cs+ and Sr2+ and the incorporation of these large cations led to the decrease in crystallinity [6].
Appendix references
[3] M.M. Woolfson. An Introduction to X-ray Crystallography, 1970.
[4] A. Guiner. X-ray Diffraction in Crystals, Imperfect Crystals and Amorphous Bodies, 1963.
[5] Y. Wang, J. Sun, T. Munir, B. Jia, A. Gul. Various morphologies of clinoptilolites synthesized in alcohol-solvent hydrothermal system and their selective adsorption of CH4 and N2. Micro. Meso. Mater. 2021, 323, 111235.
[6] I. Graça, D. Chadwick. NH4-exchanged zeolites: Unexpected catalysts for cyclohexane selective oxidation. Micro. Meso. Mater. 2019, 294, 109873.
The above descriptions were added in lines 124-134 of the revised manuscript, as follow:
In Figure 1A, no major structural damage was observed after ion-exchange of CP, as intensities of the main diffraction peaks for the HEU structures remained similar [29]. (020), (200) and (131) peaks intensities of NH4-CP became stronger, but the intensities of other characteristic diffraction peaks remained nearly constant compared with that of (Na, K)-CP. Furthermore, the d values of (020), (200) and (131) were expanded to 0.8927, 0.7907 and 0.3976 nm, respectively (as shown in Table S1). These results indicate the neatly arrangement of NH4+ on the crystal planes. However, the decrease in intensity of the diffraction peaks was observed for the Cs+ and Sr2+ exchanged CPs (Figure 1A and Figure S1B), which could be attributed to the high X-ray absorption of Cs+ and Sr2+ and the incorporation of these large cations led to the decrease in crystallinity [30].
Comments 2-04: line 163: the IR bands are not due to bridging OH molecules. 3607 and 1640 are the bands of adsorbed water; the 2981 band is probably a C-H stretch of carbonaceous impurities.
Response 2-04: We thank the reviewer’s useful suggestions. You are right, the peaks observed at approximate 3630 cm-1 belonged to the adsorbed water molecules, and the peak located at 1638 cm-1 corresponded to deformation vibrations of adsorbed water [7]. Additionally, the peak appeared at 2981 cm-1 may be assigned to a C-H stretch of carbonaceous impurities [8]. Therefore, the FT-IR spectra of the related samples were retested (as shown in Figure S2) considering possible contamination of the samples.
Appendix references
[7] E. A. Paukshtis, M. A. Yaranova, I. S. Batueva, B. S. Bal'zhinimaev. A FTIR study of silanol nests over mesoporous silicate materials. Micro. Meso. Mater. 2019, 288, 109582.
[8] C. F. Lien, Y. F. Lin, Y. S. Lin, M. T. Chen, J. L. Lin. FTIR Study of Adsorption and Surface Reactions of N(CH3)3 on TiO2. J. Phys. Chem. B 2005, 109, 10962-10968.
The above description was added in lines 164-166 of the revised manuscript, as follow:
The peaks observed at approximate 3630 cm-1 belonged to the adsorbed water molecules, and the peak located at 1638 cm-1 corresponded to deformation vibrations of adsorbed water [34].
Figure S2. FT-IR spectra of various CPs.
Comments 2-05: line 228-9: it is the hydration radius of Li; for Cs it is the size of the cation even without hydration water.
Response 2-05: We thank the reviewer’s useful suggestions. On the basis of Kennedy et al. demonstration [9], the radius of the exchanged cations is listed in Table 1.
Table 1. Collections of the radius of the exchanged cations
|
Cation |
Effective Ionic Radius (nm) |
Hydrated Radius (nm) |
|
Li+ |
0.076 |
0.145 |
|
Ca2+ |
0.100 |
0.180 |
|
Sr2+ |
0.118 |
0.200 |
|
Cs+ |
0.167 |
0.260 |
As can be seen in Figure 2 and Figure S8, Li+ ions show the smallest effective ionic radius and hydrated radius. The exchange equilibrium of Li+ needed more time than that of Ca2+, the probable reason is that Li+ ions occupy more exchanged sites in the HEU structures. Meanwhile, it is likely that coordination of the Li+ cations with K+ results in occupying positions within the channels [9]. The ionic radius of Cs+ is larger than that of Ca2+, which makes it difficult to enter the microporous channels and achieve exchange equilibrium. Additionally, the most exchange times were needed to reach equilibrium for Sr2+, which may be related to the exchanged sites in CP [10].
Appendix references
[9] D.A. Kennedy, F.H. Tezel. Cation exchange modification of clinoptilolite- Screening analysis for potential equilibrium and kinetic adsorption separations involving methane, nitrogen, and carbon dioxide. Micro. Meso. Mater. 2018, 262, 235-250.
[10] N. Lihareva1, O. Petrov, L. Dimowa1, Y. Tzvetanova1, I. Piroeva, F. Ublekov, A. Nikolov. Ion exchange of Cs+ and Sr2+ by natural clinoptilolite from bi‑cationicsolutions and XRD control of their structural positioning. Journal of Radioanal. Nucl. Chem. 2020, 323, 1093-1102.
The above descriptions were added in lines 230-239 of the revised manuscript, as follow:
However, Figure 2 and Figure S8 demonstrate that the exchange procedure of Cs+ and Li+ needed more times to reach the exchange equilibriums than that of Ca2+. The main reasons could be interpreted as follows: On the hand, the ionic radius of Cs+ (0.26 nm) is larger than that of Ca2+ (0.18 nm), which makes it difficult to enter the microporous channels and achieve exchange equilibrium [44]. On the other hand, Li+ ions occupy more exchanged sites in the HEU structures, although the effective ionic radius (0.076 nm) and hydrated radius (0.145 nm) of Li+ are lower than that of Ca2+. Meanwhile, it is likely that coordination of the Li+ with K+ results in occupying positions within the channels [44]. Additionally, the most exchange times were needed to reach equilibrium for Sr2+, which may be related to the exchanged sites in CP [42].
Comments 2-06: eq. 1 is strange to me and needs explanation. Is it a general expressions valid for exchange of univalent cations and divalent cations?
(1)
where ZA, ZB, respectively, are the charges of the cations A and B and the symbols ‘c’ and ‘s’ refer to the clinoptilolite (CP) and solution phases, respectively.
The equivalent fractions of the cations in the aqueous and solid phases are described by the following equations:
(2)
(3)
Appendix reference
[11] A. Alshameri, A. Ibrahim, A. M. Assabri, X. Lei, H. Wang, C. Yan. The investigation into the ammonium removal performance of Yemeni natural zeolite: Modification, ion exchange mechanism, and thermodynamics. Powder Technol. 2014, 258, 20-31.
Therefore, the equation was revised into:
Accordingly, the thermodynamic parameters were recalculated and shown in Table 2, Figure 3 and Figure S10.
|
(A) |
(B) |
Figure 3. Relationships between ln k and (RT) - 1×104 (A), as well as ln ka and (RT) - 1×104 (B)
(A)
Figure S10. Relationship between ln ka and (RT) - 1×104, a s well as ln k and (RT) - 1×104 for Li-x-CP (A)
Table 2. Summaries for ΔrGθ m (kJ/mol), ΔrSθ m (J/mol⋅k), ΔrHθ m (kJ/mol), and Ea (kJ) values of various cationic exchange process with NH4-CP.
|
Samples |
Exchange times |
313 K |
333 K |
353 K |
ΔrHθ m |
Ea |
|||
|
ΔrGθ m |
ΔrSθ m |
ΔrGθ m |
ΔrSθ m |
ΔrGθ m |
ΔrSθ m |
||||
|
Li-x-CP |
2 |
-4.603 |
30.59 |
-4.897 |
29.64 |
-5.796 |
30.50 |
4.976 |
6.220 |
|
3 |
-3.226 |
22.69 |
-3.602 |
22.45 |
-4.138 |
22.70 |
3.878 |
7.349 |
|
|
4 |
-2.934 |
23.16 |
-3.513 |
23.50 |
-3.857 |
23.14 |
4.316 |
7.839 |
|
|
5 |
-2.390 |
14.39 |
-2.631 |
14.25 |
-2.968 |
14.40 |
2.117 |
8.380 |
|
|
6 |
-2.244 |
15.61 |
-2.609 |
15.76 |
-2.863 |
15.59 |
2.643 |
9.319 |
|
|
Cs-x-CP |
2 |
-3.479 |
14.99 |
-3.706 |
14.77 |
-4.081 |
14.99 |
1.214 |
3.504 |
|
3 |
-2.749 |
30.08 |
-3.482 |
30.48 |
-3.949 |
30.08 |
6.672 |
6.406 |
|
|
4 |
-2.694 |
23.12 |
-3.105 |
22.97 |
-3.623 |
23.13 |
4.547 |
8.230 |
|
|
5 |
-2.442 |
12.39 |
-2.581 |
12.07 |
-2.823 |
12.07 |
1.439 |
10.373 |
|
|
6 |
-2.273 |
13.22 |
-2.497 |
13.10 |
-2.804 |
13.23 |
1.867 |
11.370 |
|
|
Ca-x-CP |
2 |
-1.593 |
8.641 |
-1.749 |
8.591 |
-1.868 |
8.441 |
1.113 |
5.082 |
|
3 |
-1.217 |
14.16 |
-1.406 |
13.87 |
-1.578 |
13.57 |
3.216 |
8.866 |
|
|
4 |
-1.114 |
9.663 |
-1.225 |
9.416 |
-1.380 |
9.322 |
1.912 |
9.283 |
|
|
Sr-x-CP |
2 |
-1.431 |
24.14 |
-1.629 |
23.28 |
-2.009 |
23.04 |
6.128 |
5.308 |
|
4 |
-0.968 |
11.71 |
-1.116 |
11.46 |
-1.265 |
11.23 |
2.701 |
6.409 |
|
|
6 |
-0.803 |
9.976 |
-0.912 |
9.703 |
-1.055 |
9.560 |
2.321 |
9.836 |
|
|
8 |
-0.548 |
5.109 |
-0.602 |
4.965 |
-0.687 |
4.924 |
1.052 |
12.604 |
|
Comments 2-07: The constant I is in my opinion the entropy term.
Response 2-07: Thank for the reviewer’s suggestions. According to the literature [12], the standard enthalpy of ion exchange can be evaluated as follow:
where I is a constant.
Appendix reference
[12] A. Alshameri, A. Ibrahim, A. M. Assabri, X. Lei, H. Wang, C. Yan. The investigation into the ammonium removal performance of Yemeni natural zeolite: Modification, ion exchange mechanism, and thermodynamics. Powder Technol. 2014, 258, 20-31.
Comments 2-08: line 284: improved temperature: what is it?
Response 2-08: Thank for the reviewer’s useful suggestions. As shown in Table 2, the negative ΔrGθ m values suggest that the ion exchange processes were feasible and spontaneous in nature. For all cation-exchanged clinoptilolites, the ΔrGθ m values were decreased with the increase of ion exchange temperature from 313 to 353 K, implying reasonable rising temperature is beneficial to promote the exchange process. Pandey et al. also found a similar phenomenon [13].
Appendix references
[13] S. Pandey, E. Fosso-Kankeu, M.J. Spiro, F. Waanders, N. Kumar, S.S. Ray, J. Kim, M. Kang. Equilibrium, kinetic, and thermodynamic studies of lead ion adsorption from mine wastewater onto MoS2-clinoptilolite composite. Powder Technol. 2014, 258, 20-31.
The above descriptions were added in lines 288-295 of the revised manuscript, as follow:
As shown in Table 2, the negative ΔrGθ m values suggest that the ion exchange processes were feasible and spontaneous in nature. For all cation-exchanged CPs, the ΔrGθ m values were decreased with the increase of ion exchange temperature from 313 to 353 K. For example, ΔrGθ m value for Cs-6-CP (or Ca-4-CP) was around -0.025 (or -1.114) kJ/mol at 313 K, -0.181 (-1.225) kJ/mol at 333 K, and -0.416 (-0.581) kJ/mol. These results imply that the reasonable rising temperature is beneficial to promote the exchange process, due to accelerating the diffusion rate of exchangeable cations at high temperature. Pandey et al. also found a similar phenomenon [28].

Reviewer 3 Report
This paper deals with the adsorption property of various clinoptilolite samples. The authors highlight the unique adsorption property of NMC-2. I agree well with significance of the paper, but there are several points to be considered prior to publication.
Major Point
- With regard to Figure 3 and Table 2, how many times did you carried out the experiments for each plot.
- I cannot understand well about the increase in the entropy through the cation exchange process. The present result is not consistent with those reported by Tarasevich and Argun et al. The explanation for this point should be added more clearly.
- With regard to Figures 5 and 6, the explanation and discussion on the difference in adsorption heat in (Na, K)-CP and Ca-4-CP should be added.
Minor Point
- Irregular lines are present in Tables 2 and 3.
Author Response
Reviewer #3:
This paper deals with the adsorption property of various clinoptilolite samples. The authors highlight the unique adsorption property of NMC-2. I agree well with significance of the paper, but there are several points to be considered prior to publication.
Response: Thank you very much for the positive comments.
Comments 3-01: With regard to Figure 3 and Table 2, how many times did you carried out the experiments for each plot.
Response 3-01: Thank the referee for useful suggestion. In Figure 3 and Table 2, the experiments for each plot were repeated for three times and the obtained data were averaged. The above description was added in the Experiment section.
Comments 3-02: I cannot understand well about the increase in the entropy through the cation exchange process. The present result is not consistent with those reported by Tarasevich and Argun et al. The explanation for this point should be added more clearly.
Response 3-02: Thank referee for useful suggestion.
Argun et al reported [14] the removal of Ni2+ ion from aqueous solution using clinoptilolite as an adsorbent, which was mainly an Ni2+-adsorption process. In contrast, our work investigated the cation-exchange behaviors with the exchangeable cations of clinoptilolite, therefore, the purpose of washing with deionized water in the cation-exchange experiments was to remove the cations adsorbed on clinoptilolite.
As demonstrated by Barros et al. [15] and Tarasevich et al. [16], the cation adsorption process is usually exothermic, whereas the cation exchange behavior could be endothermic and entropy increasing process.
In our work, the exchange entropy involved the change in entropy of the solid and liquid phase of the system, therefore, the contributions to ΔrSθ m may arise from changes in water-cation environments, particularly, between the aqueous solution and clinoptilolites during the exchange process [17]. Additionally, an overall negative or positive change in entropy was also linked to the enthalpy values of the exchange process.
Appendix references
[14] M.E. Argun. Use of clinoptilolite for the removal of nickel ions from water: Kinetics and thermodynamics. J. Hazard. Mater. 2008, 150, 587-595.
[15] M.A.S.D. Barros, P.A. Arroyo. Thermodynamics of the Exchange Processes between K+, Ca2+ and Cr3+ in Zeolite NaA. Adsorption. 2004, 10, 227-235.
[16] Y. I. Tarasevich, V. E. Polyakov. Ion-exchange equilibria and exchange heats on clinoptilolite involving singly-charged cations. Theor. Exp. Chem. 1996, 32, 276-280.
[17] S. Tangkawanit, K. Rangsriwatananon, A. Dyer. Ion exchange of Cu2+, Ni2+, Pb2+ and Zn2+ in analcime (ANA) synthesized from Thai perlite. Micro. Meso. Mater. 2005, 79, 171-175.
The above descriptions were added in lines 304-319 of the revised manuscript, as follow:
As shown in Table 2, the exchange process of various cations belonged to an entropy enhancement, similar results were also described by Pandey et al. [28]. However, the present results were not consistent with those reported by Argun et al. [27] and Tarasevich et al. [49]. Particularly, Argun et al. [27] found the negative ΔrSθ m values during the removal of Ni2+ ion from aqueous solution using clinoptilolite as an adsorbent, which was mainly an Ni2+-adsorption process. In contrast, our work investigated the cation-exchange behaviors with the exchangeable cations of clinoptilolite, therefore, the purpose of washing with deionized water in the cation-exchange experiments was to remove the cations adsorbed on clinoptilolite. As demonstrated by Barros et al. [48], the cation adsorption process is usually exothermic, whereas the cation exchange behavior could be endothermic and entropy increasing process. In our work, the exchange entropy involved the change in entropy of the solid and liquid phase of the system, therefore, the contributions to ΔrSθ m may arise from changes in water-cation environments, particularly, between the aqueous solution and clinoptilolites during the exchange process [49]. Additionally, an overall negative or positive change in entropy was also linked to the enthalpy values of the exchange process.
Comments 3-03: With regard to Figures 5 and 6, the explanation and discussion on the difference in adsorption heat in (Na, K)-CP and Ca-4-CP should be added.
Response 3-03: Thank the referee for useful suggestion.
As shown in Figure 6A and C, the adsorption heat of CO2, CH4 and N2 showed a similar tendency with the increase of absorbed amount for (Na, K)-CP and Ca-4-CP. However, the adsorption heat of CO2 and CH4 increased, but that of N2 decreased with the increase of their absorbed amount for Ca-4-CP, implying a stronger adsorption affinity between Ca-4-CP and CO2 (or CH4), which is related to the exchanged cationic property and the polarizability of adsorbate molecules [18, 19].
Appendix references
[18] F. Mani, J. A. Sawada, S. M. Kuznicki. A comparison of the adsorptive behavior of ETS-10, 13X, and highly siliceous ZSM-5. Micro. Meso. Mater. 2015, 214, 32-40.
[19] M. Vosoughi, H. Maghsoudi, S. Gharedaghi. Ion-exchanged ETS-10 adsorbents for CO2/CH4 separation: IAST assisted comparison of performance with other zeolites. J Nat Gas Sci Eng. 2021, 88, 103862.
The above descriptions were added in lines 411-416 of the revised manuscript, as follow:
As shown in Figure 6A and C, the adsorption heat of CO2, CH4 and N2 showed a similar tendency with the increase of absorbed amount for (Na, K)-CP and Ca-4-CP. However, the adsorption heat of CO2 and CH4 increased, but that of N2 decreased with the increase of their absorbed amount for Ca-4-CP, implying a stronger adsorption affinity between Ca-4-CP and CO2 (or CH4), which is related to the exchanged cationic property and the polarizability of adsorbate molecules [55, 56].
Comments 3-04: Irregular lines are present in Tables 2 and 3.
Response 3-04: Thank the referee for useful suggestion. Irregular lines were deleted in the revised Tables 2 and 3. Additionally, irregular lines were also deleted in the revised Tables S1.

Round 2
Reviewer 2 Report
I have no remarks on the revised manuscript
Reviewer 3 Report
The authors characterized well the adsorption properties of cation-exchanged clinoptilolites and I agree well with the findings of the present paper.